# Development and validation of a clinical score for identifying patients with high risk of latent autoimmune adult diabetes (LADA): The LADA primary care-protocol study

**Pilar Vich-Pérez**[1]*, **Juan Carlos Abánades-Herranz**[2], **Gustavo Mora-Navarro**[3], **Ángela María Carrasco-Sayalero**[4], **Miguel Ángel Salinero-Fort**[5], **Ignacio Sevilla-Machuca**[6], **Mar Sanz-Pascual**[7], **Cristina Álvarez Hernández-Cañizares**[8], **Carmen de Burgos-Lunar**[9], **LADA-PC Research Consortium**[¶]

1 Internal Medicine Specialist, Member of the MADIABETES Research Group, Principal Investigator, Los Alpes Health Center, Madrid, Spain, 2 Member of the MADIABETES Research Group, Monóvar Health Center, Madrid, Spain, 3 Member of the RedGDPS Foundation, Los Alpes Health Center, Madrid, Spain, 4 Specialist in Clinical Immunology, Ramon y Cajal University Hospital, Madrid, Spain, 5 Head of the Knowledge Management Area of the Madrid Health Council, Scientific Director of the MADIABETES Research Group, Madrid, Spain, 6 Los Alpes Health Center, Madrid, Spain, 7 Member of the MADIABETES Research Group, Aquitania Health Center, Madrid, Spain, 8 Canillejas Health Center, Madrid, Spain, 9 Member of the MADIABETES Research Group, Specialist in Epidemiology and Public Health, San Carlos Clinical Hospital, Madrid, Spain

¶ Membership of LADA-PC Research Consortium is provided in the Acknowledgments.
* pilar.vich@salud.madrid.org

## Abstract

### Background

Latent autoimmune diabetes in adults (LADA) is a type of diabetes mellitus showing overlapping characteristics between type 1 Diabetes Mellitus and type 2 Diabetes Mellitus (T2DM), and autoimmunity against insulin-producing pancreatic cells. For its diagnosis, at least one type of anti-pancreatic islet antibody (GADAb is the most common) is required. Many authors recommend performing this measure in all newly diagnosed patients with DM, but it is not possible in Primary Health Care (PHC) due to its high cost.

Currently, a relevant proportion of patients diagnosed as T2DM could be LADA. Confusing LADA with T2DM has clinical and safety implications, given its different therapeutic approach.

The main objective of the study is to develop and validate a clinical score for identifying adult patients with DM at high risk of LADA in PHC.

### Methods

This is an observational, descriptive, cross-sectional study carried out in Primary Care Health Centers with a centralized laboratory. All people over 30 years of age diagnosed with diabetes within a minimum of 6 months and a maximum of 4 years before the start of the study will be recruited. Individuals will be recruited by consecutive sampling. The study variables will be obtained through clinical interviews, physical examinations, and electronic

**Data Availability Statement:** All relevant data from this study will be made available upon study completion.

**Funding:** This study has been funded by Instituto de Salud Carlos III (ISCIII) through the project "PI19/01569" and co-funded by the European Union.

**Competing interests:** The authors have declared that no competing interests exist.

medical records. The following variables will be recorded: those related to Diabetes Mellitus, sociodemographic, anthropometric, lifestyle habits, laboratory parameters, presence of comorbidities, additional treatments, personal or family autoimmune disorders, self-perceived health status, Fourlanos criteria, and LADA diagnosis (as main variable) according to current criteria.

## Discussion

The study will provide an effective method for identifying patients at increased risk of LADA and, therefore, candidates for antibody testing. However, a slight participation bias is to be expected. Differences between participants and non-participants will be studied to quantify this potential bias.

## Introduction

Latent autoimmune diabetes in adults (LADA) is a type of diabetes mellitus (DM) showing overlapping characteristics between type 1 DM (T1DM) and type 2 DM (T2DM), and autoimmunity against insulin-producing pancreatic cells [1,2]. There is no agreement about if LADA is a different entity, or a subtype of T1DM with slower destruction of insulin-producing cells. The prevalence of LADA varies according to the studies and geographical locations, with range between 3 to 12% of all patients with DM. In the UKPDS3 study, the prevalence of antibodies against pancreatic islets (anti-glutamic acid decarboxylase-Anti-GAD-) was 12% of all patients with T2DM, although the prevalence in patients diagnosed with DM before the age of 35 years could reach 34% [3].

These data indicate that LADA would be the second most common type of DM, after T2DM, and the most common type of autoimmune diabetes in adults. In the European multicenter Action LADA study, which included 6,000 patients with recently diagnosed DM, a positivity of anti-pancreatic islet antibodies of 9.7% was documented, showing differences between southern and northern Europe countries, with a lower prevalence in the southern countries [4]. Other studies such as CARDS [5] show similar global European prevalence. Studies carried out in different countries [6] show variable results (United Kingdom, 12% [3]; Italy, 4.5% [7]; Korea, 5.1% [8]; China, 5.9% [9]; Finland, 9.3% [1]; the United Arab Emirates, 2.6% [10]; Canada-USA-Europe (2004), 4.2% [11]; Bulgaria, 10.1% [12]; Sardinia, 5% [13]) (Table 1).

In Spain, the prevalence of Anti GAD65 antibodies in a T2DM population in southern Spain was 3.7% [15]. In another study, not designed for this purpose, only patients with the LADA1 subtype were identified, with at least two anti-islet antibodies and associated overweight; a prevalence of 2.9% was found (95% CI: 0.6–8.3%) [16]. The differences found in the prevalence of positivity to Anti-GAD in the different studies could be due to the sensitivity and specificity of the reagents used, variable inclusion criteria, differences in lifestyle and ethnic characteristics.

T2DM is the predominant form of adult diabetes mellitus, and most of the new diagnoses made in Spain take place in Primary Health Care (PHC). Furthermore, most of the population with DM is controlled in PHC and patient data are coded by professionals in electronic medical records (EMR), which is a useful tool for research studies.

In the Community of Madrid PHC, EMR has been available since 1999. The analytical parameters of the patients are automatically recorded in the EMR from the central laboratory

**Table 1. LADA prevalence in different studies.**

| Authors/study | Year | Country | Type of antibody used | Prevalence |
|---|---|---|---|---|
| Turner. UKPDS 25 [3] | 1997 | UK | Anti-GAD, ICA | 12% |
| Tuomi [1] | 1999 | Finland | Anti-GAD | 9.3% |
| Zinman [11] | 2004 | Canada, USA, Europe | Anti-GAD | 4.2% |
| Buzzetti [7] | 2007 | Italy | Anti-GAD, IA-2A | 4.5% |
| Lee [8] | 2009 | Korea | Anti-GAD, IA-2A | 5.1% |
| Maioli [13] | 2010 | Sardinia (Italy) | Anti-GAD | 5% |
| Zhou [9] | 2013 | China | Anti-GAD | 5.9% |
| Hawa. Action LADA 7 [4] | 2013 | Europe | Anti-GAD, IA-2A, ZnT8A | 9.7% |
| Hawa. CARDS study cohort [5] | 2014 | Europe | Anti-GAD, IA-2A, ZnT8A | 7.1% |
| Maddaloni [10] | 2015 | Arab Emirates | Anti-GAD, IA-2A | 2.6% |
| Zaharieva [12] | 2017 | Bulgaria | Anti-GAD, IA-2A, ZnT8A | 10.1% |

Modified from Pozzilli P et al. [14].

(although they can also be entered manually if they come from external laboratories). The disease diagnoses are logged by professionals using the International Classification of Primary Care, 2nd edition (ICPC-2), some of them being validated, through studies carried out for this purpose, such as hypertension or DM [17].

Currently, a relevant proportion of patients diagnosed in PHC consultations as T2DM could be LADA. The reasons for this incorrect coding would be inadequate knowledge of this entity by professionals, difficulty in requesting anti-pancreatic islet antibodies from PHC office, or the routine management of these patients as T2DM. In addition, the diagnosis criteria in the onset of the disease may be masked over the years, because, although LADA is generally manifested in younger and slimmer patients with lower prevalence of hypertension, both the body mass index, as blood pressure, increases with age and it is possible that LADA patients, in their middle ages of life, show indistinguishable phenotypes from T2DM.

Confusing LADA with T2DM has clinical and safety implications, given its different therapeutic approach [18,19]. Although there are currently no defined therapeutic guidelines for this disease, sodium and glucose cotransporter 2 (SGLT2) inhibitors should not be used in LADA patients due to the increased risk of diabetic ketoacidosis [20–22]. On the other hand, although Sulfonylureas are effective as blood glucose lowers, they could increase the immune response and early deterioration of residual pancreatic function, so they are not recommended in these patients [18]. Metformin could be used in obese or insulin resistance LADA patients, but it does not slow the progression towards insulin dependence [19]. Dipeptidyl Peptidase 4 (DPP-4) inhibitors and glucagon-like peptide-1 (GLP-1) analogs could have some benefit in these patients, although more studies are required in this regard [19,23–29].

A recent study suggests that the addition of sitagliptin to insulin in LADA patients preserves pancreatic beta-cell function better than insulin alone [30]. Rosiglitazone was studied in LADA patients and seemed to offer some benefit in preserving the pancreatic insulin reserve [19]; however, cardiovascular risks associated with this drug advised its withdrawal from the market in the past decade. Insulin appears as the most recommended treatment in LADA patients because it is the one that best preserves the functionality of the pancreatic beta cell [19,31–33].

For the diagnosis of LADA, the Immunology of Diabetes Society [14] proposed the following criteria: age of onset over 30 years; the presence of at least one type of anti-pancreatic islet antibody (glutamic acid decarboxylase autoantibodies -GADAs, anti-GAD or GADAb-,

insulin autoantibodies -IAA-, protein tyrosine phosphatase IA-2 -IA-2A-, islet-specific zinc transporter isoform 8 -ZnT8- autoantibodies), and insulin independence the first six months after diagnosis of the disease.

In LADA, the most frequently found antibodies are GADAs, and many authors recommend performing this measure in all newly diagnosed diabetic patients [18]. However, the costs involved in following this recommendation would be high and hardly acceptable in PHC.

Recently published studies suggested the relevance of applying clinical scores that make LADA diagnosis more likely, identifying patients for whom it is cost effective to request antibodies [18]. Back in 2006, Fourlanos and collaborators proposed a clinical tool to identify adults at high risk of LADA [34,35] with five criteria (age <50 years at diagnosis of DM, body mass index at diagnosis less than 25 Kg/m$^2$, symptoms hyperglycemia at diagnosis, personal history of autoimmune disease linked to HLA DR3 / DQ2 or DR4 / DQ8 and family history of autoimmune disease linked to HLA DR3 / DQ2 or DR4 / DQ8).

The presence of two or more of these criteria would identify nine out of ten patients with LADA. The presence of one or no criteria would make the diagnosis of LADA highly unlikely.

The Fourlanos study had a retrospective phase in which patients with known LADA were included in the interviewing process. They could have introduced a bias (greater probability of remembering their weight, height, autoimmune diseases given that they are aware of having a case of atypical diabetes). In addition, the methodology for weighting the ORs obtained in the multivariate logistic regression analysis to obtain the score was not the most usually used. These circumstances, and the fact that no subsequent studies have been carried out in other geographical areas to validate score, would make necessary, from our point of view, to locally validate it, and to test the possibility of including other variables that are more frequent in patients with LADA (for example, HDL cholesterol which could be higher in LADA than in T2DM) [4].

A recent, retrospective Spanish study of 193 patients seen in endocrinology consultations of the public system of the Community of Madrid [36] reflected a diagnostic delay of LADA of 3.5 years, average age at diagnosis of 49 years, time of insulin-independence of 12 months, presence of other autoimmune pathologies in 57% of cases and a poor glycemic control despite intensive insulin therapy. Although blood pressure and lipids were controlled, these populations were often overweight, with a low incidence of macroangiopathic complications. Although this work provides interesting information on these patients, the study population is not comparable to that of PHC because, in this area, the representation of patients with T2DM comprises a broader and more heterogeneous group of people, the patients have a longer follow-up and it is where we believe there is a greater proportion of patients with unidentified LADA and confused with T2DM.

The purpose of this study is to develop a clinical score that allows identifying patients at high risk of LADA who may benefit from requesting anti-GAD antibodies, making the diagnosis of this disease in PHC more efficient. Likewise, these patients will be characterized, comparing them with patients with T2DM and the frequency of this entity will be estimated.

## Materials and methods

### Research objectives

Develop and validate a clinical score to identify adult diabetics at high risk of LADA in PHC.

Estimate the frequency of LADA in newly diagnosed adult diabetic patients in PHC.

Describe the characteristics of these patients (No LADA / LADA) in the study population.

**Project Design**: Observational, descriptive, multicenter, and cross-sectional study.

**Scope**: Multicenter. Madrid Urban Health Centers with a centralized laboratory into the Ramón y Cajal University Hospital in Madrid. The initial pilot project will include Los Alpes, Aquitania, and Monóvar Health Centers.

**Study Population**: All diabetic patients older than 30 years diagnosed during the last 4 years (and who have been diagnosed for at least 6 months) belonging to several urban health centers in Madrid, identified by consulting the Health Management Information Systems of the Primary Care Assistance of Madrid. 1,660 patients will be needed for the development of the score.

**Inclusion Criteria**: patients over 30 years of age who have been diagnosed with DM during the last 4 years and who have been diagnosed for at least 6 months.

**Exclusion Criteria**: homebound subjects, gestational diabetes, patients who do not wish to participate and those who cannot collaborate because of physical and/or mental disabilities.

**Classification of the Spanish Agency for Medicines and Health Products** (month/day/year; 11/16/2017) Observational Study No Post-authorization.

**Expected Start of the Study**: 01/02/2022 for the pilot project.

**Estimated enrollment date of the last patient**: 03/31/2023.

**Sample and sampling**: Consecutive sampling until reaching the 1,660 patients necessary for the study. The total number of patients who meet the inclusion criteria will be included and those with any exclusion criteria will be excluded. For the pilot study, at least the first 400 patients of 30 places will be included, from three selected health centers.

## Default sample size for general project

a. **For developing the clinical score**: Assuming that the clinical score to be developed has a sensitivity of 90%, for a desired precision of 5%, a confidence interval of 95% and an expected prevalence of LADA of 10%, 1,383 patients diagnosed with DM in the last 3 years will be required.

b. **For estimating the proportion of patients diagnosed with DM in the last 3 years who are LADA**: For an expected proportion of 10%, a precision of 2%, a confidence interval of 95% and considering a finite population of 820,000 patients with diabetes in the Community of Madrid, 863 patients will be required.

c. **For obtaining a clinical score with an adequate number of predictors**: the sample size will be determined with Freeman's rule, number of cases = 10 x (k + 1), where the number of cases is the number of patients who have developed LADA, k is the number of possible predictor variables. If a prevalence of LADA = 10% is assumed and 6 predictors are to be used, 700 patients will be necessary, and 800 patients will be required for 7. Therefore, 1,383 patients will be needed to carry out the study, to achieve a safe size for the proposed objectives. If the number of patients with LADA in the estimated study period does not reach the expected value (1,383 patients), the sample size will be expanded with the participation of other centers or the study until this value is achieved. Considering that 20% of the recruited patients could drop out of the study, the number of patients that will have to be recruited summarizes 1,660.

## Duration and development of the study

The expected length of the activities is estimated at 3 years for the complete study and 6 months for the piloting to ensure technical, administrative and logistic feasibility of the full-scale study and to avoid interfering with the normal development of the consultation and to take advantage of some of the analytical determinations of the patients.

At the inclusion visit, it will be verified that the patient meets the diagnostic criteria for DM, based on ADA criteria: 1. Cardinal symptoms + blood glucose $\geq$ 200 mg/dl. 2. HbA1c

≥6.5% on at least two occasions. 3. Two blood glucose values ≥126 mg/dl. 4. Glycemia 2 hours after an oral glucose tolerance test with 75 g of glucose ≥200 mg/dl on two occasions. Likewise, it will be verified that the patient has been diagnosed at an age greater than 30 years, does not present physical or mental disabilities that prevent him from participating, and is not gestational diabetes. Once the patients have read the patient information sheet and signed the informed consent, they will be recalled to carry out a complete analysis if they have not had one recorded for the last six months (hemogram, complete biochemical analysis, HbA1c, and microalbuminuria). All patients will be asked for Anti-GAD and antithyroid antibodies (the latter in case they have not been performed in the last year). Booking of the analysis will be forced to match the next one scheduled according to the current protocol in T2DM.

At the study visit, which will take place between 7–30 days after recruitment, the patient will be called for a consultation to obtain all the study variables and inform him of the analysis results.

The variables that make up the Fourlanos clinical score [34] and the rest of the study variables will be collected.

Some of the required variables may be obtained from the electronic medical record, such as previous pathologies, medication, personal history, and some analytical and anthropometric data from the last six months; others will require the presence of the patient if it is not registered in the medical record, or are not recent, such as height, weight, blood pressure, waist circumference, and current analytical data that include anti-pancreatic islet cell antibodies and anti-thyroid antibodies. Likewise, a questionnaire on life habits and sociodemographic aspects will be carried out.

To facilitate a more effortless follow-up of participants, each researcher will generate a list of patients under their care. They will write down the numerical code obtained after inclusion in the electronic data collection notebook. This code will allow the link with the electronic medical records when necessary. The Fig 1 shows the study flow chart.

**Management of losses**: It is expected that a percentage of the patients will not wish to participate, will not go to the health center, will be assisted in other centers, or may die during the time of the study. The family doctor or nurse will try to recruit the maximum number of patients, informing them that the additional time in scheduled consultation to complete data on their history and disease will allow a better knowledge of their type of DM. In anticipation of an approximate 20% of losses, the sample size has been increased to 1,660 patients since it is intended to reach the number of 138 patients with LADA (among newly diagnosed adult diabetics), to avoid losing the score accuracy. If necessary, the sample size can be expanded until this value is reached.

**Variables** (included in the electronic data collection notebook (eDCN) in standardized format https://lada-ap.es/acceso.html. See Supporting information for more details.

Main variable (S1 Table): LADA criteria according to the Immunology of Diabetes Society [14].

Sociodemographic variables (S2–S5 Tables): Age, sex, education level, family, and social support.

Anthropometric variables (S6 Table): Height (cm), Weight (Kg), BMI (Kg/m$^2$), Waist circumference (cm).

Clinical variables (S7–S14 Tables): Date of diagnosis of DM, Diagnostic method of DM, Symptoms of hyperglycemia at diagnosis (polyuria, polydipsia, weight loss), Life habits (tobacco, alcohol, physical activity [37], physical involvement work, Mediterranean diet [38].

Laboratory parameters (S15 Table): Blood glucose value at diagnosis (mg/dl and mmol/L), HbA1c value at diagnosis (% and mmol/mol), HbA1c value in the last 6 months (% and mmol/mol), Albumin/creatinine ratio, Ketonuria at some point in evolution, Total cholesterol

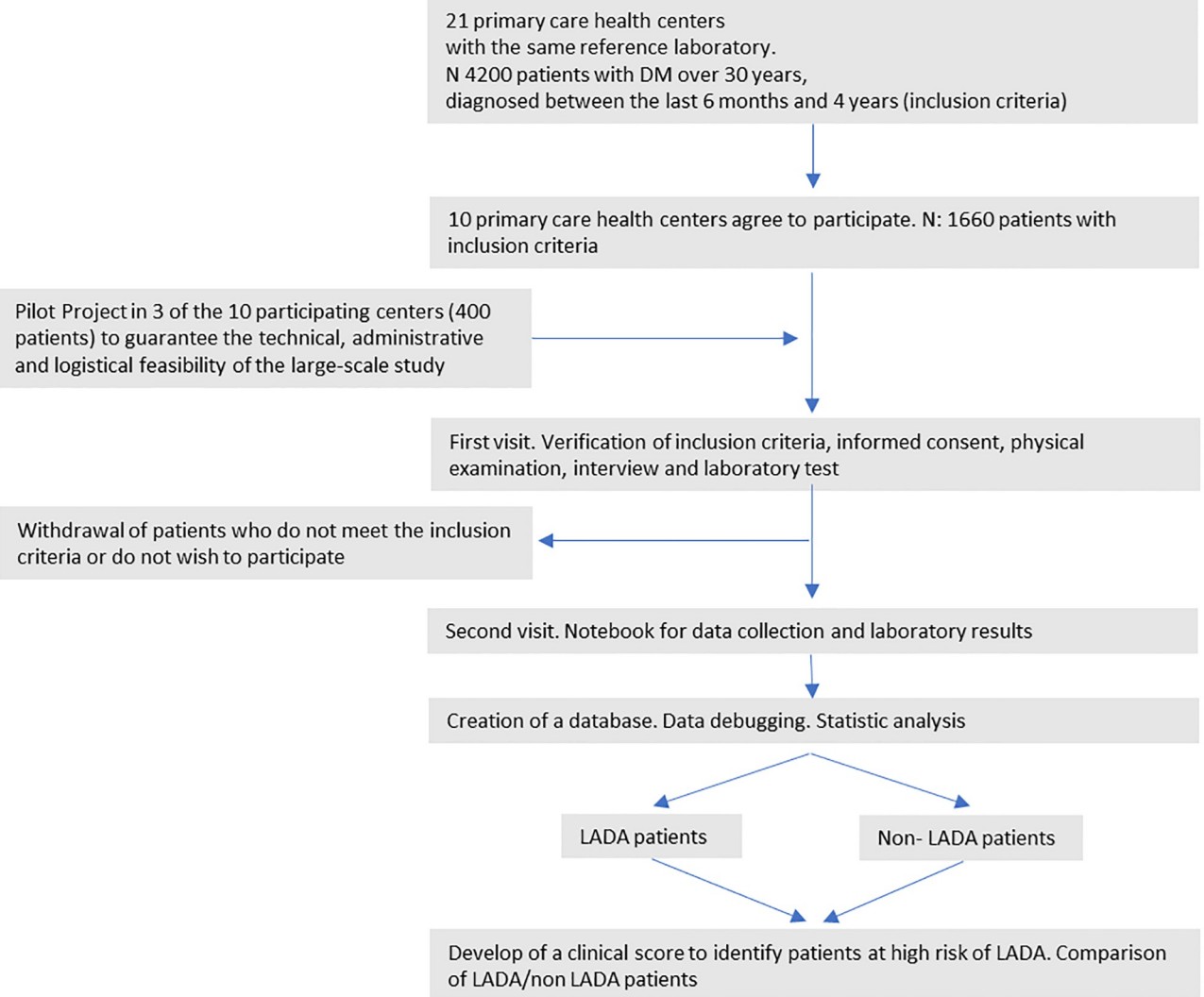

**Fig 1. Flow chart of the LADA primary care protocol study.**

(mg/dl), HDL cholesterol (mg/dl), LDL cholesterol (mg/dl), Triglycerides (mg/dl), Anti GAD 65 antibodies: (+/- and titer -IU/mL-), Anti-peroxidase antibodies: (+/- and titer-IU/mL-), Antithyroglobulin antibodies: (+/- and titer-IU/mL-).

Treatment for diabetes mellitus (S16 Table): lifestyle performances, Metformin, Sulfonyl-ureas, Dipeptydil-peptidase-4 inhibitors, Pioglitazone, Repaglinide, SGLT-2 inhibitors, GLP-1 analogs, Insulin.

Other treatments (S17 Table): angiotensin converting enzyme inhibitors (ACEI), angiotensin receptor blockers (ARBs), Non-ACEI/ARBs antihypertensives, Lipid-lowering drugs, anxiolytics and hypnotics drugs, antidepressants drugs, antipsychotics drugs, antiplatelet drugs, anticoagulant drugs, Analgesics, Anti-tumor drugs, Immunomodulator drugs, corticosteroids.

Comorbidities (S18 Table): Microalbuminuria, Chronic Kidney Disease, Diabetic retinopathy, Diabetic neuropathy, arterial hypertension, hypercholesterolemia, hypertriglyceridemia, metabolic syndrome, ischemic heart disease, peripheral arterial disease, cerebrovascular disease, atrial fibrillation, heart failure, COPD, asthma, mood disorder, psychotic disorder.

Metabolic syndrome (S19 Table)

Personal history of autoimmune disorder linked to HLA DR3 / DQ2 or DR4 / DQ8 (S20 Table): Autoimmune thyroid disease, pernicious anemia, celiac disease, Addison's disease, vitiligo, rheumatoid arthritis, autoimmune hepatitis, T1DM.

First or second-degree family history (parents, children, siblings, grandparents, great-uncles, and nephews) of an autoimmune disorder linked to HLA DR3 / DQ2 or DR4 / DQ8 (S21 Table): Autoimmune thyroid disease, pernicious anemia, celiac disease, Addison's disease, vitiligo, rheumatoid arthritis, autoimmune hepatitis, T1DM, and number of affected relatives.

Perceived health status (S22 Table):

Fourlanos criteria (LADA Clinical Score by Fourlanos et al. S23 Table) [34].

## Statistical analysis

First, the descriptive statistics of the demographic and clinical characteristics of the patients included in the study will be carried out, stratifying between patients with and without LADA according to the criteria of the Immunology of Diabetes Society [14]. The estimation of patients with LADA will be made with their 95% confidence interval. For comparison between subgroups of quantitative variables, the student's t test or the non-parametric Mann-Whitney U test will be applied in the case of non-normal distributions. For the comparison of qualitative variables, $\chi$ non-asymptotic Monte Carlo methods and the exact test will be used. The interpretation of the tables will be carried out using the corrected standardized residuals.

Finally, to find predictive factors for LADA, a binary logistic regression model will be performed after a previous univariate analysis, which will identify the independent variables related to LADA that will be entered in the multivariate model for the final selection.

The evaluation of the validity of the model will be carried out using the Shrinkage method, whose objective is to improve the predictions from the original logistic regression model. The discriminatory capacity of the model will also be determined by calculating Harrell's C statistics and measures of the discriminatory power of the models in the development and validation samples. This analysis will be performed on the original models and on the models adjusted by a uniform Shrinkage factor [39]. The difference of the 2 Harrell's C statistics must be less than 0.10 to consider that the model created has discriminatory validity and can be considered a highly reliable model.

The models obtained will be calibrated by studying the concordance between the observed results and those predicted by the generated models. The calibration of the model will be carried out using a goodness of fit test ($\chi^2$ statistic and Hosmer-Lemeshow statistic) [40], calculating the calibration slopes and calibration graphs. Given the dichotomous nature of the response variable, the calibration of the predicted risk probabilities will be studied by deciles.

To develop the clinical score for the study, the score for each variable will first be estimated using univariate logistic regression. Next, the six variables most strongly associated with LADA will be selected to be included in the multivariate logistic regression model. Third, to derive the scores assigned to each variable from the new score, the beta coefficients obtained in the multivariate model will be multiplied by 10. This procedure has been used in the development of similar scores [41–43]. The total score of the clinical score of the study will be the sum of these coefficients. Higher scores will correspond to an increased risk of LADA.

The diagnostic performance of the model will be calculated in terms of sensitivity, specificity, and predictive values, in the total population studied and it will be replicated in the population $\geq 30$ years. Statistical analysis will be performed with SPSS Statistics for Windows, version 21.0 (IBM Corp, Armonk, New York, USA) and MedCalc for Windows, version 15.8 (MedCalc Software bvba, Ostend, Belgium).

**Ethical-Legal and Regulatory Aspects**: Positive approval of the Regional Research and Medicines Ethics Committee and the Central Research Commission of the Primary Care Management of Madrid has been obtained. All data gathered will be treated confidentially. In the electronic data collection notebook, these will appear dissociated, so that it is not possible to identify the patients by people outside the project (compliance with Organic Law 3/2018, of December 5, on Protection of Personal Data and guarantee of digital rights). Furthermore, Helsinki Standards (2013) will be met.

Written informed consent will be requested from the patient and confidentiality commitment from the researchers. The study does not imply any change in the usual clinical practice.

**Promoters**: LADA-AP is promoted by the Madrid Primary Care Assistance Management. It has the collaboration of the Foundation for Biosanitary Research and Innovation in Primary Care and the Immunology Service of the Ramón y Cajal University Hospital.

**Website**: https://lada-ap.es/.

## Discussion

The study will provide an effective method for identifying patients at increased risk of LADA and, therefore, candidates for antibody testing.

## Limitations of the study

This is a multicenter study where many patients and the collaboration of many clinical researchers is necessary. To standardize the collection of information, a meeting will take place in each participating center prior to data collection. An initial pilot in three health centers will be carried out to detect problems and identify areas for improvement. A slight participation bias is to be expected. Differences between participants and non-participants will be studied to quantify this potential bias. The study population includes all patients with a recent diagnosis of DM and the recruitment will be carried out by their reference professionals who are expected to achieve maximum participation. There may also be a memory bias in the personal or family history, due to the patients not remembering some of them. In a recent study in which six risk scores were analyzed in general practice, De la Iglesia et al. [44] highlighted the lack of data, especially concerning family history. As a possible solution, family doctors will receive training for effective anamnesis to reduce this bias.

To ensure the nonexistence of a population selection bias, we will conduct a sensitivity analysis to compare the LADA frequency between neighborhood deprivation index quartiles.

Lastly, the professionals may show a greater interest in obtaining the recruitment of patients whom they consider to be more likely to have LADA-type diabetes. Unfortunately, this fact could overestimate the true frequency of LADA-type diabetes in newly diagnosed patients.

## Supporting information

**S1 Table. Main variable.** LADA criteria according to the Immunology of Diabetes Society [14].
(DOCX)

**S2 Table. Sociodemographic variables: Age.**
(DOCX)

**S3 Table. Sociodemographic variables: Sex.**
(DOCX)

**S4 Table. Sociodemographic variables: Education level.**
(DOCX)

**S5 Table. Sociodemographic variables: Family and social support.**
(DOCX)

**S6 Table. Anthropometric variables.**
(DOCX)

**S7 Table. Clinical variables: Date of diagnosis of DM.**
(DOCX)

**S8 Table. Clinical variables: Diagnostic method of DM.**
(DOCX)

**S9 Table. Clinical variables: Symptoms of hyperglycemia at diagnosis.**
(DOCX)

**S10 Table. Clinical variables.** Life habits: Tobacco.
(DOCX)

**S11 Table. Clinical variables.** Life habits: Alcohol.
(DOCX)

**S12 Table. Clinical variables.** Life habits: Physical activity level [37].
(DOCX)

**S13 Table. Clinical variables.** Life habits: Physical involvement work [37].
(DOCX)

**S14 Table. Clinical variables.** Life habits: Mediterranean diet [38].
(DOCX)

**S15 Table. Laboratory parameters.**
(DOCX)

**S16 Table. Treatment for diabetes mellitus.**
(DOCX)

**S17 Table. Other treatments.**
(DOCX)

**S18 Table. Comorbidities.**
(DOCX)

**S19 Table. Metabolic syndrome.**
(DOCX)

**S20 Table. Personal history of autoimmune disorder linked to HLA DR3 / DQ2 or DR4 / DQ8.**
(DOCX)

**S21 Table. First or second-degree family history (parents, children, siblings, grandparents, great-uncles, and nephews) of an autoimmune disorder linked to HLA DR3 / DQ2 or DR4 / DQ8.**
(DOCX)

**S22 Table. Perceived health status: Excellent / Very good / Good / Fair / Poor.**
(DOCX)

**S23 Table. Fourlanos criteria (LADA Clinical Score by Fourlanos et al [35].**
(DOCX)

## Acknowledgments

We would like to thank Teresa Sanz Cuesta for her invaluable methodological support that was key to obtaining the funding.

LADA-PC Research Consortium: María Victoria García Espinosa (a1), Ignacio Sevilla Machuca (a1), Margarita Puerto Rodríguez (a1), Gema Izquierdo Enríquez (a1), Belén Vicente Mata (a1), Oscar Sánchez López (a1), Pilar Vich Pérez (a1), Begoña Brusint Olivares (a1), Isabel Prieto Checa (a1), Blanca Jerez Basurco (a1) Ana Isabel Moreno Gómez (a1), Sara Ascensión Pérez Medina (a1), Raquel Cabral Rodríguez (a1), María Dolores Martín Álvarez (a1), Paula Regueiro Toribio (a1), Alejandro Valenzuela Luque (a1), Carlos Settanni Gomis (a1), Elena Díaz Crespo (a1), Cecilia Rufino Cano (a1), Irene Moratinos Recuenco (a1), Maria José Guereña Tomás (a1), Nuria Campos Campos (a1), Raul Coleto Gutierrez (a1), Susana Madero Velazquez (a1), María Fernández de Paul (a1), Laura Vazquez López (a1), Almudena Cardenas de Miguel (a1), Pablo Betrián González (a1), Encarnación Ayuso Gil (b1), Paloma Martínez Amigo (b1), Miriam Castro Benito (b1), Lourdes Botanes Peñafiel (b1), Irene Duque Rebollo (b1), Blanca Tobar Lomas (b1), Ana María García Ortega (b1), Eva Sánchez García (b1), María Oreja de Vega (b1), Mar Sanz Pascual (a2), Marta Lor Leandro Pascual (a2), Cristina Mencia Valle (a2), Nieves Reyes Fernández (a2), Margarita Berzal Rosende (a2), Juan Carlos Platero Burgos (a2), Yunier Bidot González (a2), Yolanda De La Fuente Cortés (a2), Cristina Gómez Macho (b2), Beatriz Cáceres Sánchez (b2), M Carmen Rodríguez Romero (b2), Carlos Alberto Ortega Morán (b2), María Peña De Diego (b2), Ángeles Rodríguez Martín (b2), Rebeca Rodríguez Martínez (b2), Alejandra Cobo Mena (b2), Beatriz Ponce Moreno (b2), Juan Carlos Abánades Herranz (a3), Nuria Pertierra Galindo (a3), Sagrario Muñoz-Quirós Aliaga (a3), Yolanda Canellas Criado (a3), Beatriz Ríos Alonso (a3), Sara García Cabrera (a3), José Ignacio Vicente Díez (a3), Eva María Piqué Prado (b3), Almudena María Asensio de la Cruz (b3), Elsa Burgos Costalago (b3), Marta Nieto Gualda (b3), Cristina Santos Álvarez (a4), M Ángeles Rodriguez Sierra (a4), Alberto Serrano López Hazas (a4), M Teresa Galán Gutierrez (a4), Almudena Uranga Gómez (a4), Rosario Marta Ruiz Giardin (a4), Maria de la O Gracia Moliner (a4), Silvia Medrano Sanz (a4), Miriam Goicoechea García (a4), Aranzazu Alonso Leonardo (b4), M Teresa Cuenca Blanco (b4), M Gemma Ferrero García (b4), Javier Torcal López (b4), Vanesa Bustos Ruiz (b4), Ana María Sobrado de Vicente (a5), Elisabet Hurtado Ortega (a5), Alberto Curiel Blanco (a5), Ignacio Ortiz Mouliaa (a5), Ana Maria Herranz Torrubiano, M Reyes Ramírez Arrizabalaga (a5), Celia Casas Mena (a5), Concepción García Zarza (b5), Margarita Ortas González, Inmaculada Cuevas López (b5), María José Campos Rodríguez (b5), Nerea Gago Moreno (b5), Margarita Herrero Delgado (a6), Rafael Alonso Roca (a6), Mar Asenjo Calvo (a6), Ricardo Benito Fernández (a6), María Campos López-Carrión (a6), Luis Carrascal García (a6), M Carmen Castillo López (a6), Elena Castresana Martín de las Mulas (a6), Esther Ercilla Gorrichategui (a6), Mercedes Fernández Quesada (a6), Francisco Ayala López (a6), Ana Elena Jiménez García (a6), Sara Medina Muñoz (a6), Dolores Molero Portolés (a6), Natividad Ortega Inclán (a6), Carmen Pascual Díez (a6), Elena Pejenaute Labari (a6), Arancha Pérez Medina (a6), Mercedes Ricote Belinchón (a6), María Felizardo de Gouveia (a6), Jorge Rodríguez Reguera (a6), Gracia Elena Rogles Muñoz (a6), Mercedes Rojo Tardón (a6), Josefa Sánchez Viedma (a6), Maite Sánchez Villares Rodríguez (a6), Esperanza Villar

Coloma (a6), Ana Jiménez Arroyo (a6), Virginia Antolín Díaz (b6), Jenny Raquel Bejarano Baldeón (b6), Elena Calvo García (b6), Lidia Sánchez Hernández (b6), Ana M. Castro García (b6), M Jesús Cordero Padilla (b6), Sara Criado Jorge (b6), Elena Aranda Shaw (b6), Lorena Espinosa Rubio (b6), Esperanza Fernández Sanz (b6), Tirso Galiano Arroyo (b6), Virginia García Campo (b6), Isabel García del Río (b6), Raquel García de la Rubia Usero (b6), Verónica Garrido Herrera (b6), Leticia Valero Sánchez (b6), Ana María López Herrera (b6), Beatriz Mallavibarrena Ramírez (b6), Elvira Muñoz Millán (b6), Laura Tejedor Posadas (b6), Aránzazu Ordóñez Vinuesa (b6), Almudena Pazos González (b6), Juan Antonio Prado García Moreno (b6), Ana Ruiz Jiménez Alfaro (b6), Carolina Sánchez Stephan (b6), Pilar Serrano Muñoz (b6), Lorena Fernández Pérez (b6), Magda Lucy Vargas Reyes (b6), Freddy Velazco Cárdenas (b6), Isabel García del Río (b6), Cristina Álvarez Hernández-Cañizares (a7), Nieves Domínguez Agüero (a7), Cristina Garcimartin del Rey (a7), Cristina Gónzalez Sanchez (a7), Cristina Escudero Lafont (a7), Amparo Pozo Teruel (a7), Antonio García López (a7), Maria Luisa Rodriguez (a7), Emilia Eguía Hormigos (a7), Aurea Redondo Sendino (a7), Monserrat Gómez Cuñarro (a7), Laura Rodríguez Alonso (b7), Blanca Vallejo Campo (b7), Mar Amor Moncada (b7), Alberto Redondo Gómez (b7), Gloria Leyva Vera (b7), Elena Martin Herrero (b7), Elena Montes Algarra (b7), M Luisa Romero Molina (b7), Esmeralda Pulido López (b7), Almudena Toro Herrero (b7), MPilar Narro Pérez (b7), Julián Díaz Sánchez (a8), María Domínguez Paniagua (a8), M Pilar Serrano Simarro (a8), German Reviriego Jaén (a8), Cristina Montero García (a8), M Inmaculada González García (a8), Irma Ramos Gutiérrez (a8), Adrián Rocha Alcubilla (a8), Norma Fernández Alonso (b8), M Teresa de la Jara Leal (b8), Laura Cristino Fuentes (b8), Laura García Villoslada (b8), Margarita Camarero Shelly (b8), Marina Prado Calvete (b8), M Guadalupe García Martín (b8), Sofía Blanca Patón (b8), M Teresa Gómez Martínez (b8), M Gracia González Rodriguez (b8).

*(a) Family doctor*, *(b) Nurse*, *(1) Los Alpes Health Center*, *(2) Aquitania Health Center*, *(3) Monóvar Health Center*, *(4) García Noblejas Health Center*, *(5) Alameda de Osuna Health Center*, *(6) Mar Baltico Health Center*, *(7) Canillejas Health Center*, *(8) Barajas Health Center*

## Author Contributions

**Conceptualization:** Pilar Vich-Pérez, Juan Carlos Abánades-Herranz, Miguel Ángel Salinero-Fort.

**Data curation:** Miguel Ángel Salinero-Fort, Carmen de Burgos-Lunar.

**Formal analysis:** Miguel Ángel Salinero-Fort, Carmen de Burgos-Lunar.

**Funding acquisition:** Pilar Vich-Pérez.

**Investigation:** Pilar Vich-Pérez, Juan Carlos Abánades-Herranz, Gustavo Mora-Navarro, Ángela María Carrasco-Sayalero, Ignacio Sevilla-Machuca, Mar Sanz-Pascual, Cristina Álvarez Hernández-Cañizares.

**Methodology:** Pilar Vich-Pérez, Juan Carlos Abánades-Herranz, Miguel Ángel Salinero-Fort.

**Project administration:** Juan Carlos Abánades-Herranz.

**Supervision:** Pilar Vich-Pérez, Juan Carlos Abánades-Herranz, Gustavo Mora-Navarro.

**Validation:** Pilar Vich-Pérez, Carmen de Burgos-Lunar.

**Writing – original draft:** Pilar Vich-Pérez, Gustavo Mora-Navarro.

**Writing – review & editing:** Juan Carlos Abánades-Herranz, Miguel Ángel Salinero-Fort.

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
