## [Decision Letter · Decision Letter 0]

28 Sep 2022

PONE-D-22-01490DEVELOPMENT AND VALIDATION OF A CLINICAL SCORE FOR IDENTIFYING PATIENTS WITH HIGH RISK OF LATENT AUTOIMMUNE ADULT DIABETES (LADA): THE LADA PRIMARY CARE-PROTOCOL STUDY.PLOS ONE

Dear Dr. Vich-Perez,

Thank you for submitting your manuscript to PLOS ONE. After careful consideration, we feel that it has merit but does not fully meet PLOS ONE’s publication criteria as it currently stands. Therefore, we invite you to submit a revised version of the manuscript that addresses the points raised during the review process.

We look forward to receiving your revised manuscript.

Kind regards,

Vijayaprakash Suppiah, PhD

Academic Editor

PLOS ONE

Journal Requirements:

2. One of the noted authors is a group or consortium ( LADA-PC Research Consortium). In addition to naming the author group, please list the individual authors and affiliations within this group in the acknowledgments section of your manuscript. Please also indicate clearly a lead author for this group along with a contact email address

4. Please include your tables as part of your main manuscript and remove the individual files. Please note that supplementary tables (should remain/ be uploaded) as separate "supporting information" files

Reviewers' comments:

Reviewer's Responses to Questions

**Comments to the Author**

1. Does the manuscript provide a valid rationale for the proposed study, with clearly identified and justified research questions?

Reviewer #1: Yes

Reviewer #2: Yes

Reviewer #3: Yes

2. Is the protocol technically sound and planned in a manner that will lead to a meaningful outcome and allow testing the stated hypotheses?

Reviewer #1: Yes

Reviewer #2: Partly

Reviewer #3: Yes

3. Is the methodology feasible and described in sufficient detail to allow the work to be replicable?

Reviewer #1: Yes

Reviewer #2: Yes

Reviewer #3: Yes

4. Have the authors described where all data underlying the findings will be made available when the study is complete?

Reviewer #1: Yes

Reviewer #2: Yes

Reviewer #3: Yes

5. Is the manuscript presented in an intelligible fashion and written in standard English?

Reviewer #1: Yes

Reviewer #2: Yes

Reviewer #3: Yes

6. Review Comments to the Author

You may also provide optional suggestions and comments to authors that they might find helpful in planning their study.

Reviewer #1: Important study for primary clinical practice, but far implications on LADA patients

management

Otherwise, Fourlanos criteria must be upgraded and improved.

Reviewer #2: Thank you very much for inviting me to review this interesting manuscript.

This manuscript describes the development and validation of a clinical score that might help to identify patients at high risk of developing LADA in primary care (The LADA primary care study) in addition to estimate the frequency of LADA in newly diagnosed adult patients in Madrid and to describe the characteristics of these patients ( LADA/non LADA). It includes a pilot project (as a proof of concept) followed by the main study.

It is important to diagnose LADA at presentation so the correct therapeutic measures can be implemented while minimizing the risk of developing DKA and a simple, clinic score tool might facilitate this process. The prevalence of LADA in Madrid - and in general in Spain- has not been fully described in the literature hence the proposed objectives are indeed valid.

However, a number of questions require some clarification and these are listed below alongside with some comments/suggestions:

1. Although most of the proposed protocol can be found within the document, this has been presented in a slightly disjointed manner, making quite challenging to smoothly follow it. For example, in the "discussion" the following paragraph " each researcher will generate a list of patients [...] the usual clinical practice" will be better placed in methods. The flow chart does not clearly differentiate between the pilot and the main study nor the two groups of patients that will be compared ( LADA and non LADA). This is also rather unclear in the main body of the document.

2. This has been described as an observational, cross sectional, multicenter study. The study subjects are 30 years old (or older) who have been diagnosed with type 2 diabetes in the last 4 years (and diagnosed for at least 6 months). Is the data collected mainly retrospectively using the electronic data collection notebook (eDCN) or will it also include prospective data (and it so with what purpose and how often)? Where will be this prospective data recorded? This should be clearer in the main body of the paper.

3. Is the length of the study 3 or 4 years, please? Both durations have been mentioned within the document. Please, also clarify the estimated enrollment date for the last patient ( 03/31/2023) since this is just above one year from the start of the study. If this is correct, please clarify why the study will be extended for another 3 years.

4. How would the incidence of LADA be estimated in new patients if this is mainly a retrospective study?

5. Is the population served by the 3 pilot centers and the 7 remaining centers that agreed to participate in the study representative of the overall population in Madrid? Is there any variation in ethnicity or deprivation indexes for example that could potentially cause bias? And if so, how this will be addressed, please?

6. How do you propose to recruit the patients for the study? Could most of the data be extracted just by using the electronic data collection notebook (eDCN) rather than by interviewing? How will the patients be consented?

7. Exclusion criteria should include gestational diabetes and use of steroids at the time of diagnosis.

8. Sample size calculation for developing the clinical score: this has been calculated assuming a 10% prevalence of LADA however this prevalence seems to be variable depending on geographical locations. Northern European countries appear to have a higher prevalence than Southern European countries. Although Action LADA 7 (reference 4) suggested a prevalence of 9.7%, the 9 participating countries were predominantly from North/Central Europe with an estimated prevalence of around 10% whereas in Italy (another participating country, also reference 6) it has been reported to be as low as 4.5%. Action LADA 7 also selected patients from both primary and secondary care centers which might bias the data further. Reference 14 reports a prevalence of 3.7% in South Spain. Could this variation in prevalence impact in the size of the cohort required for the study and if so, how will this be addressed, please?

9. Please, kindly report HbA1c concentration in IFCC-standardized units (mmol/mol) in addition to DCCT units (%). It will be also helpful if blood glucose values could be reported in mmol/L.

10. In the flow chart, it reads “First visit. Inclusion criteria. Informed consent. Exam, interview and lab test”. What type of exam (examination?) will be conducted?. Please, if abbreviations are used, clarify what they mean.

11. I read with interest that anti peroxidase antibodies and anti-thyroglobulin antibodies will be tested as part of the protocol. Anti- thyroid peroxidase antibodies (anti-TPO antibodies) are a marker of thyroid autoimmunity and anti-thyroglobulin antibodies are primarily used as a long term tool to monitor patients with thyroid cancer. The second will have no added value in the proposed study. If the former is used to check for additional autoimmunity why this has been limited to the thyroid gland and not extended to additional autoimmune processes elsewhere? ( Coeliac, Addison, Pernicious anemia and so on). Would it be more informative if anti tyrosine phosphatase like insulinoma ( IA-2) and Zinc transporter 8 ( Zn-T8) antibodies are checked instead?

12. Please, clarify the reasons behind using the criteria of the Immunology of Diabetes Society to stratify patients with and without LADA rather than the Fourlanos criteria.

13. Could you provide some of the references used to describe the validity of the model, the discriminatory capacity of the model and the calibration of the model, please?

14. Could you anticipate any additional bias that could affect the study design?

15. I strongly recommend to find the support of a scientific English editor. Although the document is written in standard English, there are multiple typos and inaccurate/incorrect wording. Some examples to illustrate this are:

- instead of "more profitable"= cost effective

- instead of "practically exclude the disease"= highly unlikely

- instead of " to validate it in our environment"= to locally validate it

- instead of "geographic"= geographical areas

- instead of "losses of patients"= patient drops out

- instead of " they will be summoned"= they will be recalled or invited

- instead of "inclusion clearance"= following recruitment

- instead of "size" ( kindly check in anthropometric variables)= height

- instead of "conceptualization"= study conception

- instead of "data curation"= data collection

- instead of " formal analysis"= statistical analysis

- instead of "methodology"= study design

- patients are not LADA nor gestational diabetes nor T2DM but have LADA, gestational diabetes or T2DM

- it is unclear what "systematic of 18 determinations of biochemistry "means.

- in the flow chart it is mentioned "patients with T2DM over 30 years"= do you mean the patients have T2DM for more than 30 years or that the patients are 30 years old or older?

- usually up to 3 references can be added in the document independently (19, 20, 21), if more than 3 consecutive references are mentioned a different format is preferred (18, 22-28)

Overall, good concept and effort with good supporting references however, major changes are required, especially in the description of the pilot, main study, material and methods and flow chart.

Reviewer #3: 1. Page 11 Paragraph 7: Authors mentioned that “the fact that no subsequent studies have been carried out in other geographic areas to validate score, would make necessary, from our point of view, to validate it in our environment and to test the possibility of including other variables that are more frequent in patients with LADA (for example, HDL cholesterol that is higher in LADA than in T2DM)” -

Is HDL cholesterol always high in all LADA patients than T2DM? How is this a specific parameter to score and screen for LADA? Also, in the subsequent paragraph reference 37 it is mentioned that most of the patients screened were well controlled with lipids.

2. How much does the prior treatment with drugs for DM /other drugs listed in S17 table affect the score?

3. How is the final score arrived at?

4. Table S17 covers a range of drugs covered for different conditions. The effect of each of these drugs can be varied and affect the scoring. Whether this can make a bias in deriving the score?

7. PLOS authors have the option to publish the peer review history of their article (what does this mean?). If published, this will include your full peer review and any attached files.

Reviewer #1: **Yes: **Zoran Gluvic, MD, PhD

Reviewer #2: No

Reviewer #3: No

---

## [Author Response · Author response to Decision Letter 0]

28 Oct 2022

Dear editor,

We thank the editor and the three reviewers for their comments on our manuscript. Below is our response to each point the academic editor and reviewers raised. We hope that we satisfyingly addressed them, and that the manuscript will be now suited for publication. 

Sincerely, 

On behalf of all authors,

Pilar Vich Pérez

Reviewer #1: Important study for primary clinical practice, but far implications on LADA patients

management

Otherwise, Fourlanos criteria must be upgraded and improved.

We fully agree with the reviewer that Fourlanos criteria need to be reviewed and improved. Likewise, we believe that the identification of patients with LADA type diabetes (LADA patients) allows better therapeutic management and better results.

Reviewer #2: Thank you very much for inviting me to review this interesting manuscript.

This manuscript describes the development and validation of a clinical score that might help to identify patients at high risk of developing LADA in primary care (The LADA primary care study) in addition to estimate the frequency of LADA in newly diagnosed adult patients in Madrid and to describe the characteristics of these patients ( LADA/non LADA). It includes a pilot project (as a proof of concept) followed by the main study.

It is important to diagnose LADA at presentation so the correct therapeutic measures can be implemented while minimizing the risk of developing DKA and a simple, clinic score tool might facilitate this process. The prevalence of LADA in Madrid - and in general in Spain- has not been fully described in the literature hence the proposed objectives are indeed valid.

However, a number of questions require some clarification and these are listed below alongside with some comments/suggestions:

1. Although most of the proposed protocol can be found within the document, this has been presented in a slightly disjointed manner, making quite challenging to smoothly follow it. For example, in the "discussion" the following paragraph " each researcher will generate a list of patients [...] the usual clinical practice" will be better placed in methods. The flow chart does not clearly differentiate between the pilot and the main study nor the two groups of patients that will be compared ( LADA and non LADA). This is also rather unclear in the main body of the document.

We appreciate the reviewer's comments, and the Flow Chart has been improved. We have tried to present the document in a more joint way to make it easier to read.

Also, we have moved and improved the paragraph to the appropriate section in material and methods. Now it reads as follows:

To facilitate a more effortless follow-up of participants, each researcher will generate a list of patients under their care. They will write down the numerical code obtained after inclusion in the electronic data collection notebook. This code will allow the link with the electronic medical records when necessary.

2. This has been described as an observational, cross sectional, multicenter study. The study subjects are 30 years old (or older) who have been diagnosed with type 2 diabetes in the last 4 years (and diagnosed for at least 6 months). Is the data collected mainly retrospectively using the electronic data collection notebook (eDCN) or will it also include prospective data (and it so with what purpose and how often)? Where will be this prospective data recorded? This should be clearer in the main body of the paper.

As stated in the material and methods, it is a cross-sectional study without a second follow-up phase. The participants are selected from electronic medical records and with the following criteria: patients over 30 years of age who had been diagnosed with DM within the previous four years and a disease duration of at least six months at the recruitment time.

3. Is the length of the study 3 or 4 years, please? Both durations have been mentioned within the document. Please, also clarify the estimated enrollment date for the last patient ( 03/31/2023) since this is just above one year from the start of the study. If this is correct, please clarify why the study will be extended for another 3 years.

We only measured once each patient's demographic, anthropometric, clinical, and laboratory parameters (cross-sectional study). However, given the conditions of routine clinical practice, it will take three years to complete the analysis of the data and antibodies of the 1,660 patients. For this reason, the estimated enrollment date of the last patient is 03/31/2023.

The manuscript reads as follows:

The expected length of the activities is estimated at 3 years for the complete study and 6 months for the piloting to avoid interfering with the normal development of the consultation and to take advantage of some of the analytical determinations of the patients

4. How would the incidence of LADA be estimated in new patients if this is mainly a retrospective study?

As previously mentioned, the study is cross-sectional. Therefore, at no time will the incidence be calculated, but rather the frequency of LADA in patients diagnosed with diabetes in the last four years with a disease duration of at least six months.

The manuscript verbatim reads as follows:

Research Objectives:

Estimate the frequency of LADA in newly diagnosed adult diabetic patients in PHC.

5. Is the population served by the 3 pilot centers and the 7 remaining centers that agreed to participate in the study representative of the overall population in Madrid? Is there any variation in ethnicity or deprivation indexes for example that could potentially cause bias? And if so, how this will be addressed, please?

It is known that the clinical management of the population with diabetes that is carried out in primary care is of greater intensity in the most disadvantaged socioeconomic groups (1). Usually, glycemic control tests are performed more frequently among the most disadvantaged population, both men and women, and referrals to specialists related to diabetes grow as patients' deprivation level increases. However, once the level of need (degree of metabolic control and complications) is considered, there are no differences in referrals between social groups (2). 

On the other hand, if our objective had been to know the prevalence of LADA-type diabetes, we would have needed to have made a selection of centers and patients representative of the population. To know the frequency of LADA in patients with a recent diagnosis of diabetes in the participating centers, we did not consider it necessary to fully ensure the sample's representativeness. Nevertheless, the ten centers are representative of the community of Madrid, as has been shown in other studies (3)

However, to ensure the nonexistence of a population select bias, we will conduct a sensitivity analysis to compare the LADA frequency between neighborhood deprivation index quartiles. We have added this paragraph in the limitations section: 

To ensure the nonexistence of a population selection bias, we will conduct a sensitivity analysis to compare the LADA frequency between neighborhood deprivation index quartiles.

1. Starfield B. Primary care: an increasingly important contributor to effectiveness, equity, and efficiency of health services. SESPAS report 2012. Gac Sanit. 2012 Mar;26 Suppl 1:20-6. doi: 10.1016/j.gaceta.2011.10.009. Epub 2012 Jan 21. PMID: 22265645.

2. Amaia Bacigalupe, Santiago Esnaola, Iñaki Fraile, Josu Ibarra, Javier Urraca, Sheila Sánchez, Eduardo Millán. Desigualdades sociales en la atención a la diabetes tipo 2 en la Comarca Araba. Vitoria-Gasteiz: Departamento de Salud, Servicio de Estudios e Investigación Sanitaria 2017. Available: https://www.euskadi.eus/contenidos/informacion/equidad_en_salud/es_def/adjuntos/diabetes.pdf

3. Prevalence of diabetes mellitus and cardiovascular risk in the adult population of the Madrid Autonomous Region: 2015 PREDIMERC study. Directorate-General of Public Health, Madrid Regional Authority, 2018. Available: http://www.madrid.org/bvirtual/BVCM020168.pdf

6. How do you propose to recruit the patients for the study? Could most of the data be extracted just by using the electronic data collection notebook (eDCN) rather than by interviewing? How will the patients be consented?

Each researcher will receive a list of patients assigned to their quota who meet the pre-established inclusion criteria (age over 30 years, diagnosis of diabetes in the last four years, and time since diagnosis of at least six months). The lists will be drawn up by the team responsible for the Centralized Information System of the Public Health Service and then sent to each researcher in encrypted form. Each investigator will review these lists again to verify if the patients meet the inclusion criteria. Subsequently, they will be invited to participate in the study.

Informed consent will be requested in writing after informing the patients and when they have agreed to participate in the project.

Some of the required data may be obtained from the electronic medical record, such as previous pathologies, medication, personal history, and some analytical and anthropometric data from the last six months; others will require the presence of the patient if it is not registered in the medical record, or are not recent, such as height, weight, blood pressure, waist circumference, and current analytical data that include anti-pancreatic islet cell antibodies and anti-thyroid antibodies. Likewise, a questionnaire on life habits and sociodemographic aspects will be carried out.

The last paragraph has been added to the manuscript in the material and methods section:

Some of the required variables may be obtained from the electronic medical record, such as previous pathologies, medication, personal history, and some analytical and anthropometric data from the last six months; others will require the presence of the patient if it is not registered in the medical record, or are not recent, such as height, weight, blood pressure, waist circumference, and current analytical data that include anti-pancreatic islet cell antibodies and anti-thyroid antibodies. Likewise, a questionnaire on life habits and sociodemographic aspects will be carried out.

7. Exclusion criteria should include gestational diabetes and use of steroids at the time of diagnosis.

We agree that gestational diabetes should be an exclusion criterion, and it is noted in the protocol.

When the patient reports having consumed steroids at a time close to the diagnosis of diabetes or at the time of recruitment, they will be excluded from the study. 

8. Sample size calculation for developing the clinical score: this has been calculated assuming a 10% prevalence of LADA however this prevalence seems to be variable depending on geographical locations. Northern European countries appear to have a higher prevalence than Southern European countries. Although Action LADA 7 (reference 4) suggested a prevalence of 9.7%, the 9 participating countries were predominantly from North/Central Europe with an estimated prevalence of around 10% whereas in Italy (another participating country, also reference 6) it has been reported to be as low as 4.5%. Action LADA 7 also selected patients from both primary and secondary care centers which might bias the data further. Reference 14 reports a prevalence of 3.7% in South Spain. Could this variation in prevalence impact in the size of the cohort required for the study and if so, how will this be addressed, please?

The sample size was calculated for an estimate of LADA cases of 10%, model sensitivity of 90%, and precision of 5%. Assuming that the proportion of LADA was lower, the precision would be higher than 5%. For example: for a LADA frequency of 5% and keeping the sensitivity and the 95% CI unchanged, the precision would be 6.45% (more wide confidence interval).

However, the preliminary data from the pilot study is very close to 10%, so we estimate that the precision will not experience relevant changes.

9. Please, kindly report HbA1c concentration in IFCC-standardized units (mmol/mol) in addition to DCCT units (%). It will be also helpful if blood glucose values could be reported in mmol/L.

We agree with the reviewer to include the standardized units of measurement for Hba1c and blood glucose values.

The value of basal glycaemia at diagnosis has been included, but not other values during the evolution of the disease in the data collection notebook. We have chosen Glycated hemoglobin (HbA1c) because is considered the gold standard for monitoring and the treatment of diabetes (1)

1. Klein, K.R., Buse, J.B. The trials and tribulations of determining HbA1c targets for diabetes mellitus. Nat Rev Endocrinol 16, 717–730 (2020). https://doi.org/10.1038/s41574-020-00425-6

10. In the flow chart, it reads “First visit. Inclusion criteria. Informed consent. Exam, interview and lab test”. What type of exam (examination?) will be conducted?. Please, if abbreviations are used, clarify what they mean.

The flow chart has been modified following the recommendations of the reviewer.

Physical examination measures the patient's blood pressure, abdominal circumference, height and weight. 

11. I read with interest that anti peroxidase antibodies and anti-thyroglobulin antibodies will be tested as part of the protocol. Anti- thyroid peroxidase antibodies (anti-TPO antibodies) are a marker of thyroid autoimmunity and anti-thyroglobulin antibodies are primarily used as a long term tool to monitor patients with thyroid cancer. The second will have no added value in the proposed study. If the former is used to check for additional autoimmunity why this has been limited to the thyroid gland and not extended to additional autoimmune processes elsewhere? ( Coeliac, Addison, Pernicious anemia and so on). Would it be more informative if anti tyrosine phosphatase like insulinoma ( IA-2) and Zinc transporter 8 ( Zn-T8) antibodies are checked instead?

We agree with the reviewer that it would have been appropriate to have a complete study of autoimmunity in the patients studied. However, given that the funding budget was limited, we decided to choose autoimmune thyroid disease because it is one of the most frequent. In addition, anti-thyroid peroxidase antibodies (anti-TPO antibodies) are the most prevalent in these cases. Anti-thyroglobulin antibodies are less frequent than anti-TPO antibodies, but more than in the general population. They are frequently used for diagnosis of autoimmune thyroid diseases and the follow-up of thyroid cancer.

Finally, Anti tyrosine phosphatase-like insulinoma (IA-2) antibodies are being performed on all patients because they are included in the same KIT as anti-GAD antibodies.

12. Please, clarify the reasons behind using the criteria of the Immunology of Diabetes Society to stratify patients with and without LADA rather than the Fourlanos criteria.

The Immunology of Diabetes Society criteria are currently used for diagnosing patients with LADA, while the Fourlanos criteria set up a clinical score that allows for identifying patients with a high risk of LADA.

Fourlanos score has not been externally validated (from a statistical perspective). Therefore, one of the objectives of this research is to validate the Fourlanos score or develop a new one. 

13. Could you provide some of the references used to describe the validity of the model, the discriminatory capacity of the model and the calibration of the model, please?

As suggested by the reviewer, we add bibliographical references on the methodology used and examples of articles that have used the same method.

The evaluation of the validity of the model will be carried out using the Shrinkage method, whose objective is to improve the predictions from the original logistic regression model. The discriminatory capacity of the model will also be determined by calculating Harrell's C statistics and measures of the discriminatory power of the models in the development and validation samples. This analysis will be performed on the original models and on the models adjusted by a uniform Shrinkage factor (A). The difference of the 2 Harrell's C statistics must be less than 0.10 to consider that the model created has discriminatory validity and can be considered a highly reliable model. 

The models obtained will be calibrated by studying the concordance between the observed results and those predicted by the generated models. The calibration of the model will be carried out using a goodness of fit test (χ2 statistic and Hosmer-Lemeshow statistic) (B), calculating the calibration slopes and calibration graphs. Given the dichotomous nature of the response variable, the calibration of the predicted risk probabilities will be studied by deciles. 

To develop the clinical score for the study, the score for each variable will first be estimated using univariate logistic regression. Next, the six variables most strongly associated with LADA will be selected to be included in the multivariate logistic regression model. Third, to derive the scores assigned to each variable from the new score, the beta coefficients obtained in the multivariate model will be multiplied by 10. This procedure has been used in the development of similar scores (C-E). The total score of the clinical score of the study will be the sum of these coefficients. Higher scores will correspond to an increased risk of LADA. 

The diagnostic performance of the model will be calculated in terms of sensitivity, specificity, and predictive values, in the total population studied and it will be replicated in the population ≥ 30 years. Statistical analysis will be performed with SPSS Statistics for Windows, version 21.0 (IBM Corp, Armonk, New York, USA) and MedCalc for Windows, version 15.8 (MedCalc Software bvba, Ostend, Belgium).

(A) Steyerberg, E. W., Borsboom, G. J., van Houwelingen, H. C., Eijkemans, M. J., & Habbema, J. D. F. Validation and updating of predictive logistic regression models: a study on sample size and shrinkage. Statistics in medicine. 2004; 23(16): 2567- 2586.

(B) Hosmer DW, Hosmer T, Le Cessie S, Lemeshow S. A comparison of goodness-of-fit tests for the logistic regression model. Stat. Med. 1997; 16: 965–80.

(C) Glümer C, Carstensen B, Sandbaek A, Lauritzen T, Jørgensen T, Borch-Johnsen K. A Danish diabetes risk score for targeted screening: the Inter99 study. Diabetes Care. 2004;27(3):727-33.

(D) Wang H, Liu T, Qiu Q, Ding P, He YH, Chen WQ. A Simple risk score for identifying individuals with impaired fasting glucose in the Southern Chinese population. Int J Environ Res Public Health. 2015;12:1237-52

(E) Salinero-Fort MA, Burgos-Lunar C, Lahoz C, Mostaza JM, Abánades-Herranz JC, Laguna-Cuesta F, et al. Performance of the Finnish Diabetes Risk Score and a simplified Finnish Diabetes Risk Score in a community-Based, cross-Sectional programme for screening of undiagnosed Type 2 Diabetes Mellitus and Dysglycaemia in Madrid, Spain: The SPREDIA-2 Study. PLoS One. 2016;11(7):e0158489.

14. Could you anticipate any additional bias that could affect the study design?

In a recent study in which six risk scores were analyzed in general practice, De la Iglesia et al. (A) highlighted the lack of data, especially concerning family history. As a possible solution, family physicians will receive training for effective anamnesis to reduce this bias.

Lastly, the professionals may show a greater interest in obtaining the recruitment of patients whom they consider to be more likely to have LADA-type diabetes. Unfortunately, this fact could overestimate the true frequency of LADA-type diabetes in newly diagnosed patients

(A) de la Iglesia B, Potter JF, Poulter NR, et al. Performance of the ASSIGN cardiovascular disease risk score on a UK cohort of patients from general practice. Heart 2011;97:491-9.

15. I strongly recommend to find the support of a scientific English editor. Although the document is written in standard English, there are multiple typos and inaccurate/incorrect wording. Some examples to illustrate this are:

- instead of "more profitable"= cost effective Corrected

- instead of "practically exclude the disease"= highly unlikely Corrected

- instead of " to validate it in our environment"= to locally validate it Corrected

- instead of "geographic"= geographical areas Corrected

- instead of "losses of patients"= patient drops out Corrected 

- instead of " they will be summoned"= they will be recalled or invited Corrected

- instead of "inclusion clearance"= following recruitment Corrected

- instead of "size" (kindly check in anthropometric variables)= height Corrected

- instead of "conceptualization"= study conception Corrected

- instead of "data curation"= data collection, We prefer “data cleaning” as other authors use.

- instead of " formal analysis"= statistical analysis Corrected

- instead of "methodology"= study design Corrected

- patients are not LADA nor gestational diabetes nor T2DM but have LADA, gestational diabetes or T2DM. We can't quite understand what it is trying to tell us.

- it is unclear what "systematic of 18 determinations of biochemistry "means. These are the 18 biochemical parameters that are most requested in an analysis. Includes basic, hepatic, renal, metabolic and lipid profile. It is replaced in the text by "complete biochemical analysis"

- in the flow chart it is mentioned "patients with T2DM over 30 years"= do you mean the patients have T2DM for more than 30 years or that the patients are 30 years old or older? The flow-chart has been modified

- usually up to 3 references can be added in the document independently (19, 20, 21), if more than 3 consecutive references are mentioned a different format is preferred (18, 22-28) Corrected

Overall, good concept and effort with good supporting references however, major changes are required, especially in the description of the pilot, main study, material and methods and flow chart.

Reviewer #3: 

1. Page 11 Paragraph 7: Authors mentioned that “the fact that no subsequent studies have been carried out in other geographic areas to validate score, would make necessary, from our point of view, to validate it in our environment and to test the possibility of including other variables that are more frequent in patients with LADA (for example, HDL cholesterol that is higher in LADA than in T2DM)” -

Is HDL cholesterol always high in all LADA patients than T2DM? How is this a specific parameter to score and screen for LADA? Also, in the subsequent paragraph reference 37 it is mentioned that most of the patients screened were well controlled with lipids.

This study aims to find some of the most frequent characteristics in LADA patients compared to type 2 diabetes mellitus. It is already known that a younger age at diagnosis or a normal body mass index is more associated with LADA than with type 2 diabetes.

However, it is possible that other analytical parameters, such as high or normal HDL cholesterol, as shown by the CARDS study (1), are more frequently associated with LADA.

This research will allow us to verify it. 

(1) Hawa MI, Buchan AP, Ola T, Wun CC, DeMicco DA, Bao W, Betteridge DJ, Durrington PN, Fuller JH, Neil HA, Colhoun H, Leslie RD, Hitman GA. LADA and CARDS: a prospective study of clinical outcome in established adult-onset autoimmune diabetes. Diabetes Care. 2014 Jun;37(6):1643-9. doi: 10.2337/dc13-2383. Epub 2014 Apr 10. PMID: 24722498.

2. How much does the prior treatment with drugs for DM /other drugs listed in S17 table affect the score?

One of the implicit objectives of this study is to analyze the differences and similarities between Latent Autoimmune Adult Diabetes (LADA) and type 2 diabetes mellitus. For example, the type and level of drugs used could differ in both groups, although possibly not relevant to the development of the score.

3. How is the final score arrived at?

To develop the clinical score for the study, the score for each variable will first be estimated using univariate logistic regression. Next, the six variables most strongly associated with LADA will be selected to be included in the multivariate logistic regression model. To derive the scores assigned to each variable from the new score, the beta coefficients obtained in the multivariate model will be multiplied by 10. This procedure has been used in the development of similar scores (A-C). The total score of the clinical score of the study will be the sum of these coefficients. Higher scores will correspond to an increased risk of LADA. 

(A) Glümer C, Carstensen B, Sandbaek A, Lauritzen T, Jørgensen T, Borch-Johnsen K. A Danish diabetes risk score for targeted screening: the Inter99 study. Diabetes Care. 2004;27(3):727-33.

(B) Wang H, Liu T, Qiu Q, Ding P, He YH, Chen WQ. A Simple risk score for identifying individuals with impaired fasting glucose in the Southern Chinese population. Int J Environ Res Public Health. 2015;12:1237-52

(C) Salinero-Fort MA, Burgos-Lunar C, Lahoz C, Mostaza JM, Abánades-Herranz JC, Laguna-Cuesta F, et al. Performance of the Finnish Diabetes Risk Score and a simplified Finnish Diabetes Risk Score in a community-Based, cross-Sectional programme for screening of undiagnosed Type 2 Diabetes Mellitus and Dysglycaemia in Madrid, Spain: The SPREDIA-2 Study. PLoS One. 2016;11(7):e0158489.

The diagnostic performance of the model will be calculated in terms of sensitivity, specificity, and predictive values, in the total population studied and it will be replicated in the population ≥ 30 years. Statistical analysis will be performed with SPSS Statistics for Windows, version 21.0 (IBM Corp, Armonk, New York, USA) and MedCalc for Windows, version 15.8 (MedCalc Software bvba, Ostend, Belgium).

4. Table S17 covers a range of drugs covered for different conditions. The effect of each of these drugs can be varied and affect the scoring. Whether this can make a bias in deriving the score?

The medication of the study patients has been included as part of their characterization. 

Possibly, the patients with type 2 diabetes receive a more significant number of cardiovascular drugs than LADA patients, but it is only a hypothesis that will have to be tested. 

Additional points. 

1. The website of the LADA-AP study has been updated and referenced in the text (https://lada-ap.es/)

2. Supporting information has been modified based on reviewer suggestions.

Table S8: Units of measurement for Hb A1c in mmol/mol have been included.

Table S14: The table has been corrected because a mistake has been detected (item 3 was equal to item 5).

3. The bibliographical references have been modified adapting them to the new manuscript version.

---

## [Decision Letter · Decision Letter 1]

30 Jan 2023

DEVELOPMENT AND VALIDATION OF A CLINICAL SCORE FOR IDENTIFYING PATIENTS WITH HIGH RISK OF LATENT AUTOIMMUNE ADULT DIABETES (LADA): THE LADA PRIMARY CARE-PROTOCOL STUDY.

PONE-D-22-01490R1

Dear Dr. Vich-Perez,

We’re pleased to inform you that your manuscript has been judged scientifically suitable for publication and will be formally accepted for publication once it meets all outstanding technical requirements.

Kind regards,

Vijayaprakash Suppiah, PhD

Academic Editor

PLOS ONE

Reviewers' comments:

Reviewer's Responses to Questions

**Comments to the Author**

1. Does the manuscript provide a valid rationale for the proposed study, with clearly identified and justified research questions?

Reviewer #1: Yes

2. Is the protocol technically sound and planned in a manner that will lead to a meaningful outcome and allow testing the stated hypotheses?

Reviewer #1: Yes

3. Is the methodology feasible and described in sufficient detail to allow the work to be replicable?

Reviewer #1: Yes

4. Have the authors described where all data underlying the findings will be made available when the study is complete?

Reviewer #1: Yes

5. Is the manuscript presented in an intelligible fashion and written in standard English?

Reviewer #1: Yes

6. Review Comments to the Author

You may also provide optional suggestions and comments to authors that they might find helpful in planning their study.

Reviewer #1: The authors made an effort to improve the paper. As I already mentioned, the Fourlanos criteria must be improved regarding adaptation to current guidelines. A lot of problems encountered in the paper were in detail improved by authors.

7. PLOS authors have the option to publish the peer review history of their article (what does this mean?). If published, this will include your full peer review and any attached files.

Reviewer #1: No

---

## [Editor Report · Acceptance letter]

2 Feb 2023

PONE-D-22-01490R1 

DEVELOPMENT AND VALIDATION OF A CLINICAL SCORE FOR IDENTIFYING PATIENTS WITH HIGH RISK OF LATENT AUTOIMMUNE ADULT DIABETES (LADA): THE LADA PRIMARY CARE-PROTOCOL STUDY. 

Dear Dr. Vich-Perez:

I'm pleased to inform you that your manuscript has been deemed suitable for publication in PLOS ONE. Congratulations! Your manuscript is now with our production department. 

Kind regards, 

on behalf of

Dr. Vijayaprakash Suppiah 

Academic Editor

PLOS ONE